# Cytoskeletal Remodelling as an Achilles’ Heel for Therapy Resistance in Melanoma

**DOI:** 10.3390/cells11030518

**Published:** 2022-02-02

**Authors:** Adrian Barreno, Jose L. Orgaz

**Affiliations:** Instituto de Investigaciones Biomédicas Alberto Sols, Consejo Superior de Investigaciones Científicas (CSIC), Universidad Autónoma de Madrid, 28029 Madrid, Spain; abarreno@iib.uam.es

**Keywords:** melanoma, targeted therapy, resistance, cytoskeleton, actomyosin

## Abstract

Melanoma is an aggressive skin cancer with a poor prognosis when diagnosed late. MAPK-targeted therapies and immune checkpoint blockers benefit a subset of melanoma patients; however, acquired therapy resistance inevitably arises within a year. In addition, some patients display intrinsic (primary) resistance and never respond to therapy. There is mounting evidence that resistant cells adapt to therapy through the rewiring of cytoskeleton regulators, leading to a profound remodelling of the actomyosin cytoskeleton. Importantly, this renders therapy-resistant cells highly dependent on cytoskeletal signalling pathways for sustaining their survival under drug pressure, which becomes a vulnerability that can be exploited therapeutically. Here, we discuss the current knowledge on cytoskeletal pathways involved in mainly targeted therapy resistance and future avenues, as well as potential clinical interventions.

## 1. Melanoma and the Problem of Therapy Resistance

Melanoma is a highly aggressive cancer with a high propensity to metastasize. Early diagnosis and surgical removal of local, non-invasive tumours is curative in most cases [1]. However, late diagnosis of metastatic non-resectable melanomas carries a poor prognosis since traditional chemotherapies (dacarbazine) or immunotherapies (high dose interleukin 2) are not effective in most cases [1].

Melanomas arise from the transformation of melanocytes, which are neural-crest-derived cells specialized in the production of the pigment melanin that provides ultraviolet light protection [2]. Cutaneous melanoma originates from melanocytes in the basal layer of the epidermis, and it is the most common form of melanoma; therefore, the studies reviewed here mostly refer to cutaneous melanoma (unless otherwise stated). Other non-cutaneous melanocytes, such as those in the choroidal layer of the eye; in gastrointestinal, respiratory, and genitourinary mucosal surfaces; and in the meninges, can undergo malignant transformation, albeit at much lower frequencies [2]. Some of these non-cutaneous melanomas, such as mucosal melanomas, have an even worse prognosis [3]. Melanocytes’ neural crest ancestry is thought to contribute to the high metastatic propensity of melanomas [4,5].

Melanoma cells display enormous plasticity at epigenetic, transcriptional, translational, and post-translational levels [6]. Cutaneous melanoma cell lines and human tumours can be classified according to their gene expression profile into four phenotypes characterized by different degrees of differentiation and invasive capabilities: undifferentiated, neural crest-like, transitory, and melanocytic [7,8,9]. The majority of human melanomas from The Cancer Genome Atlas (TCGA) database can be classified as differentiated (melanocytic and transitory phenotypes) [8]. However, single-cell analysis shows that tumours can be composed of malignant cells of different phenotypes [9], strongly supporting the occurrence of phenotype interconversion [7,8,9,10]. This phenotypic plasticity (or phenotype switching) is thought to greatly contribute not only to tumour progression, dissemination, and metastasis but also, as discussed below, to therapy failure [10].

Most melanomas (≈50%) harbour mutations in the kinase BRAF (mostly BRAF^V600E^) that hyperactivate the mitogen-activated protein kinase (MAPK) pathway (RAS-BRAF-MEK-ERK) [11,12]. Therefore, treatment with small molecule inhibitors against mutant BRAF (BRAFi, vemurafenib, and dabrafenib) leads to tumour shrinkage and prolongs patient survival [13,14]. However, these responses are temporary and incomplete since some tumour cells survive under therapy and persist as a minimal residual disease, often causing relapse within a year [6,13,14]. This is the most common type of resistance to MAPK-targeted therapies (acquired resistance). In addition, around 20% of the patients never respond to therapy, displaying intrinsic or primary resistance [6,15].

In most BRAFi-resistant tumours, ERK phosphorylation is restored [16], and the restored active ERK is MEK-dependent [17,18,19]. Therefore, the combination of BRAFi with a MEK inhibitor (MEKi, trametinib) improved the overall survival (in contrast to a single BRAFi treatment) [20,21], leading to the approval of a BRAFi + MEKi combination as the standard of care for patients with metastatic melanoma [1]. However, resistance to the combination still develops [20,21,22]. In this case, most BRAFi + MEKi-resistant melanoma lines and human tumours do not restore (or poorly restore) ERK activity, with other parallel mechanisms taking over [23,24]. Other reports have shown a strong reactivation of MAPK signalling (measured as an expression of MAPK downstream targets) in most BRAFi + MEKi-resistant tumours [25]; however, phosphorylated ERK levels were not assessed here. Although not the scope of this review, it has been suggested that phosphorylated ERK levels may not be an accurate measure of MAPK pathway output [26].

A subset of around 40% of melanoma patients has impressive and long-lasting responses when treated with blocking antibodies against several immune checkpoints (PD-1, CTLA-4) [27]. These immunotherapies are aimed at disrupting those checkpoints, which function as inhibitory pathways to control immune responses and to prevent autoimmunity. Therefore, upon a checkpoint blockade, the frequently exhausted immune system would “awake,” target, and eradicate the tumour [28]. However, the majority of eligible melanoma patients do not respond to these therapies due to primary resistance [29]. This occurs through a variety of mechanisms, including oncogenic signalling, defects in antigen presentation machinery, and immunosuppression [28,29].

MAPK inhibitor (MAPKi)-resistant tumours harbour transcriptional alterations in their signalling pathways related to invasion and metastasis (epithelial-to-mesenchymal transition, TGF-β, and ECM remodelling), and immunosuppression (T cell deficiency/exhaustion and loss of antigen presentation), among other alterations [30], suggesting a cross-resistance to salvage immune checkpoint blockers (ICB). This is supported by the transcriptional analyses of tumours from ICB non-responders, which showed alterations in similar invasion/metastasis pathways [31]. In fact, some retrospective studies suggest that the progression to MAPKi is associated with inferior responses to subsequent ICB treatment [32,33,34]. Mechanistically, BRAFi-resistant tumours are cross-resistant to ICB due to ERK reactivation and subsequently enhanced MAPK pathway transcriptional output, which establishes an immunosuppressive environment with dysfunctional dendritic cells [35]. Interestingly, preclinical studies have shown that a lead-in treatment with ICB, before MAPKi, maximizes antitumour immunity and efficacy [36,37].

Therefore, despite the number of available therapies for some melanoma patients, primary and acquired therapy resistance are major problems, causing an urgent need to identify how to overcome therapy failure. In addition, therapeutic options for some non-eligible melanoma patients (for example, RAS mutant or triple BRAF/NRAS/NF1 wild-type melanoma patients) are more limited.

## 2. Adaptation and Development of Resistance to MAPK-Targeted Therapy

Intrinsic resistance to MAPK therapy is thought to arise mainly through genetic mechanisms driven by pre-existing mutations [6,38,39]. Acquired resistance can be achieved by both genetic mechanisms (de novo mutations) or nongenetic mechanisms of an early adaptive resistance that later facilitates the acquisition of secondary mutations [6,39].

During the initial adaptation to therapy, there is considerable reversible non-mutational (epigenetic and transcriptional) reprogramming [24,40], which is stably maintained (along with secondary mutations) in many completely resistant tumours [6,30,38,41]. Early drug tolerance is facilitated by an initial response phase in which MITF (melanoma oncogene and master lineage regulator) [42] is frequently induced, leading to hyperdifferentiated cells [43,44]. This is followed by the activation of stress-response pathways (JUN-JNK and ATF4, among others) [40,45,46,47,48], leading to the emergence of different, transient transcriptional states (pigmented, starved-like, invasive or de-differentiated, and neural crest stem cell-like (NCSC)), which survive as drug-tolerant persisters (DTPs) [8,39,43]. It should be noted that there is an overlap of these trajectories with the phenotypes described above for therapy-naïve melanomas [7,8]. Among these phenotypic trajectories, NCSC seem to be a major driver of therapy resistance later on, which can be delayed by targeting the retinoid X receptor (RXR) signalling [43] and a cytoskeletal regulator, focal adhesion kinase (FAK) [39] (as discussed below).

Regarding the resistant state, most resistant tumours (≈70%) display a MAPK pathway reactivation through a variety of mechanisms [16]: (1) drug targeted alterations through genetic mechanisms (BRAF amplification [49,50]) and alternative splicing [51]; (2) hyperactivation of compensatory signalling pathways (FAK [52], SRC [53], PI3K-AKT [54], and STAT3 [55]) via upregulation of receptor tyrosine kinases (AXL [56], IGFR1 [18], MET [57], EGFR [53,58], and PDGFRB [59]), including the loss of tumour suppressor PTEN [60] and activating RAS mutations [30]; (3) activation of parallel effector COT/MAP3K8 [61]; and (4) activation of downstream effectors (activating MEK mutations [49,62]). In addition, a few resistant tumours do not restore ERK phosphorylation [49,54,62,63] but seem to bypass the MAPK pathway through the activation of a more downstream oncogenic output through GPCR-cAMP-CREB signalling and amplification of MITF [64]. Furthermore, intrinsically resistant tumours seem to be in an alternative transcriptional state/phenotype, characterized by the activation of NF-kB and AXL signalling and low MITF activity [65,66]. Additional comprehensive reviews on resistance to MAPKi in melanoma can be found elsewhere [6,16,38,67].

## 3. Altered Cytoskeleton and Its Regulators in Therapy-Resistant Melanomas

Several studies using parental and BRAFi-resistant melanoma cell lines showed that some resistant sub-lines display a flatter and/or more spindle, fibroblast-like morphology with increased actin stress fibres when compared to the parental counterparts in vitro [59,68,69,70,71,72]. In some sub-lines, there are mixed subpopulations of the flat and spindle cells [59,72]. These morphological changes have also been reported during adaptation to therapy in DTP cells [24,69,73,74]. In fact, melanoma cell lines treated for only 24 h [68,69,74,75] or from 48 to 72 h [74,76,77,78] with BRAFi, MEKi, or ERKi, display spindle-shaped and/or flat morphology, regardless of the MAPK oncogenic driver (BRAF or NRAS mutant) [68].

Cell morphology is regulated by RHO GTPase signalling, which, through a variety of effectors (mostly kinases: ROCK, PAK, LIMK, ZIPK, CIT, and MRCK, among others), control the formation and activation of actin and non-muscle myosin II (NMII) filaments [79]. The actomyosin filament network, along with other components, makes up the cytoskeleton that gives shape to the cells [79]. The actomyosin cytoskeleton additionally generates the contractile force that enables cytokinesis, cell motility, migration, invasion, and force-mediated extracellular matrix (ECM) remodelling, among other fundamental processes [79,80,81].

Therefore, the morphological changes in therapy-resistant cells suggest the modulation of components of the RHO GTPase-effector-actomyosin network. In fact, the transcriptional and proteomic analyses of melanomas (cell lines and human samples) during the adaptation to MAPKi (and in resistant versus pre-treatment tumours) found a modulation of signalling pathways involved in the cytoskeletal and ECM remodelling, migration, invasion, and metastasis [24,30,40,47,68,69]. Generally, MAPKi-persistent [75,76,82] and -resistant melanomas are more invasive [53,83,84] and, with increased lung colonization ability [53], which is indicative of enhanced metastatic potential, providing further evidence of the involvement of RHO GTPase signalling and the actomyosin cytoskeleton in therapy resistance.

### 3.1. RHO GTPases

Several studies have shown the modulation of RHO GTPase expression and/or activity during adaptation to MAPKi and in MAPKi-resistant melanomas (Figure 1).

Short-term BRAFi treatment (48 h) induced an RHO GTPase switch in WM793 cells, decreasing RHOE (RND3) and increasing RHOA signalling [76]. In fact, RHOE was elevated in BRAF^V600E^ vs. non-BRAF^V600E^ melanoma cells or primary melanocytes [85].

In addition, RHOB expression was induced by a 24 h BRAFi or MEKi treatment (through JUN-dependent transcription), and RHOB inhibition increased sensitivity to BRAFi or MEKi, presumably through AKT activation [86]. In this study, mRNA levels of other atypical Rho GTPases (such as RHOJ, RHOQ, and RHOU) increased two-fold in several melanoma cell lines upon 48 h of BRAFi-treatment; however, the functional relevance was not explored.

BRAFi-resistant A375 cells displayed higher active RAC1 and RHOA levels [87]. In principle, this could be counterintuitive since therapy-naïve melanomas seem to be in either an RHOA^high^ or RAC1^high^ signalling state (which inhibit each other), driving rounded-amoeboid or elongated-mesenchymal migration, respectively [88]. However, both RHO GTPase arms cross-talk with pro-survival/proliferation signalling, given that RHOA-ROCK were shown to cooperate with pro-survival mediators (such as STAT3 [89,90] and NF-κB [91] in melanoma cells) while RAC1 activated PI3K-AKT in breast cancer cells [92]. Therefore, resistant cells that have undergone considerable stress, re-wiring their cytoskeleton may lead to the combined usage of signalling pathways that would otherwise oppose sustained survival.

RAC1^P29S^ is found in 4% of cutaneous melanoma patients and is the third most commonly mutated codon after BRAF^V600E^ and NRAS^Q61X^ [93,94]. In most cases, RAC1^P29S^ and a mutation in BRAF/NRAS/NF1 co-occur [12,95]. Using genetic mouse models, Rac1^P29S^ was found to cooperate with Braf^V600E^ in melanoma formation [95]. Rac1^P29S^ promoted dedifferentiation (via PAK and SRF/MRTF activation) and resistance to BRAFi through the suppression of apoptosis and the activation of AKT signalling and SRF/MRTF [95] (further details on MRTF are discussed below). Another study showed that RAC1^P29S^ conferred a proliferative advantage under MAPKi [77] (Figure 2). Mechanistically, this advantage was dependent on the RAC1^P29S^-mediated assembly of a dendritic actin network and lamellipodia formation, which sequestered and inactivated the tumour suppressor (and RAS inactivator) NF2/Merlin, independent of the focal adhesion and MAPK signalling. However, ECM and integrin signalling signatures were enriched in RAC1^P29S^ tumours [95]. Furthermore, the correct cell cycle and survival of RAC1^P29S^ melanoma cells were impaired by small molecule inhibitors against actin network regulators ARP2/3 [96] or PAK [97].

### 3.2. RHO GTPase Effectors

Several RHO GTPase effectors have been shown to contribute to MAPKi-resistance (Figure 1). PAK1/2 were overactivated in BRAFi- and MAPKi-resistant melanomas in vitro, most likely due to the increased expression levels of direct regulators (RAC1 and CDC42) and the PAK themselves [23]. In BRAFi-resistant melanomas, PAK phosphorylated CRAF and MEK reactivated ERK. Most MAPKi-resistant melanomas’ cell lines do not restore phosphorylated ERK (or, if very poorly) [23]. In this context, PAK activated the JNK, β-catenin, and mTOR pathways and inhibited apoptosis, thus bypassing ERK signalling [23]. Importantly, the combination of MAPKi with a PAK inhibitor overcame resistance in vivo [23].

An ectopic expression of casein kinase 2 (CKII), which can phosphorylate NMII heavy chain [98], contributed to an ERK rebound after a 48 h BRAFi treatment and increased clonogenic survival after a long-term (2 weeks) BRAFi treatment [99]. Whether CKII expression is increased in resistant cell lines and its contribution to resistance maintenance remains to be tested.

ROCK inhibition sensitized the therapy-naïve melanomas to BRAFi or ERKi [100]. Afterward, we and others found that BRAFi-resistant cells increased NMII activity (phosphorylated myosin light chain levels (p-MLC2)) [68,71,84] or total MLC2 [101]). These increased p-MLC2 levels were ROCK-dependent [68]. Consequently, the resistant cells became more vulnerable to ROCK [68,71] and NMII inhibition [68]. Mechanistically, a ROCK-NMII blockade induced a lethal reactive oxygen species, damaged DNA, reduced pro-survival STAT3-MCL1 signalling, and caused unresolved cell cycle arrest, leading to the cell’s death [68]. Thus, ROCKi combined with BRAFi overcame BRAFi resistance in vivo [68]. Importantly, ROCK-NMII also altered the tumour’s microenvironment by establishing immunosuppression (increased pro-tumourigenic CD206^+^ macrophages and FOXP3^+^ Tregs), suggesting a contribution to the cross-resistance to ICB (anti-PD-1) [68]. In fact, ROCKi, combined with an anti-PD-1 antibody, decreased the growth of the anti-PD-1-resistant melanomas through both the intrinsic tumour’s cell death and the relief of immunosuppression, without affecting tumour infiltration by CD4^+^ or CD8^+^ T cells [68]. ROCKi also reduced PD-L1 (PD-1 ligand) levels in both the tumour cells and CD206^+^ macrophages [68], which could contribute to the reduced immunosuppression and overcoming ICB resistance. STAT3 promotes PD-L1 expression [102,103]; therefore, ROCKi could reduce PD-L1 levels via a blockade of the ROCK-STAT3’s cross-talk [89]. Therapy-naïve melanomas with high NMII activity induce an NF-κB-driven immunomodulatory secretory program, including immunosuppressive cytokines (TGF-β and IL10) [91]. Therefore, therapy-resistant melanomas with a high NMII activity could be imposing the immunosuppressive environment through this secretory program.

BRAFi, MEKi, or ERKi treatment of melanoma cells for 8 to 24 h diminished p-MLC2 levels [68,75], correlating with the striking morphological changes (flat and spindle-shaped) described previously. However, p-MLC2 levels recovered to a baseline and even increased after a 48 h BRAFi treatment in A375 cells [68], which most likely explains the higher p-MLC2 levels observed later in some resistant sub-lines in vitro [68,71,84] and in tumour samples from melanoma patients after relapse [68]. In BRAFi-resistant cells, BRAFi did not affect or even increased p-MLC2 levels [68]; the latter can additionally be observed in some intrinsically resistant cells [76].

We are currently investigating these mechanisms; however, insight from published studies could provide some clues. A partial ERK reactivation during early treatment [17] could lead to a downstream amplification signalling to reactivate NMII activity. The ephrin receptor, EPHA2, was induced in BRAFi-resistant lines and contributed to their survival [83,104]. The noncanonical EPHA2 promoted a switch from RAC1 to CDC42 (and RHOA) to increase the invasion and PI3K-dependent p-MLC2 levels in therapy-naïve melanomas [84]. However, the kinetics of p-MLC2 and EPHA2 activation during BRAFi adaptation seem to be different given that EPHA2 levels, which are very low in many of the parental cell lines [83,104], were induced after only three weeks of BRAFi treatment [83]. Therefore, EPHA2 could perhaps contribute to NMII activity later in the resistant cells. An additional candidate could be RHOE; the RHOE protein’s levels decreased after a 48 h BRAFi treatment [76], while a different study showed that an RHOE knockdown increased p-MLC2 levels in therapy-naïve melanoma cells. Here, cells became flatter, with a more spread-out morphology, resembling BRAFi-treated cells [85]. RHOB could also contribute to the p-MLC2 induction upon BRAFi treatment since RHOB increased after a 48 h BRAFi treatment [86], and in certain contexts, RHOB elevated p-MLC2 levels [105,106].

ROCK and NMII signalling could additionally be increased in adapting/resistant melanomas through MITF modulation. The RNAi-mediated knockdown of MITF in therapy-naïve 501Mel cells increased the ROCK-dependent p-MLC2 levels [107] and invasion [108]. Elevated MITF levels during early BRAFi tolerance [44] could explain the reduced p-MLC2 levels in this stage. In the resistant stage, some tumours retained high MITF levels, while others displayed lower MITF levels [44]; accordingly, it would be interesting to analyse whether MITF and ROCK-p-MLC2 correlate in this setting.

Mechanotransducers YAP [109] and MRTF [110] control the actomyosin cytoskeleton and have also been involved in BRAFi resistance. YAP regulated the response to MAPKi and acted as a parallel survival pathway since there was a synthetic lethality and synergistic induction of apoptosis upon the combined YAP and MAPK inhibition [111].

YAP activation (nuclear accumulation), along with actin remodelling (stress fibres), increased after a 48 h BRAFi treatment of melanoma cells [101] and remained mostly nuclear in DTP (from 7 to 14 days of treatment) [69] and in BRAFi-resistant cells [69,71], promoting transcription [69]. YAP nuclear localization was ROCK- [71,101] and NMII-dependent [69] in the BRAFi-resistant lines. MRTF followed similar kinetics since MRTF nuclear localization increased after a 48 h BRAFi treatment and remained nuclear in some BRAFi-resistant cell lines; this regulation was RHOA- and ROCK-dependent [71,101]. RHOA-MRTF activation seems to occur predominantly in de-differentiated BRAFi-resistant cells [71]. Importantly, targeting YAP directly with verteporfin [101], indirectly through SRC inhibitor dasatinib [71], or blocking MRTF (via RNAi [101] or with an inhibitor CCG-222740 [71]) re-sensitized BRAFi-resistant cells to BRAFi. In contrast, RAC1^P29S^-mediated BRAFi resistance was significantly reversed with the MRTF inhibitor [95]. This cross-talk of RAC1^P29S^ with MRTF suggests a potential dependence on NMII for RAC1^P29S^ tumours (Figure 2). However, NMII activity levels were not assessed in this study.

### 3.3. Extrinsic Factors

The cell’s cytoskeleton senses and responds to mechanical cues from the environment through mechanotransduction [112]. In addition to the aforementioned melanoma cell-autonomous signalling, there is mounting evidence that extrinsic factors from the tumour’s microenvironment, in particular from the ECM, contribute to MAPKi resistance [40,47] (Figure 1).

Adaptation and resistance to BRAFi were accompanied by an increased ECM deposition [40,68,113], which served as a drug-protective environment [101] (Figure 1 and Figure 2). As previously discussed, collagen deposition during the early adaptation to BRAFi activated the mechanosensors YAP, MRTF, and the actomyosin cytoskeleton. This signalling, in turn, further remodelled the ECM through a positive feed-forward loop [101]. Intravital imaging of tumour xenografts confirmed that MEKi or a combined BRAFi + MEKi, increased ECM deposition and that persister cells were found adjacent to bundled collagen in vivo, which also increased the KIT-PI3K-AKT-mediated survival [114]. Furthermore, a short-term BRAFi treatment (48–72 h) induced the fibronectin expression and deposition by melanoma cells; this, in turn, protected tumour cells through α5β1-integrin-PI3K-AKT-MCL1-mediated survival [115].

Downstream of ECM receptors, such as integrins, FAK served as a signalling hub that could activate pro-survival signalling, such as MAPK or SRC-PI3K-AKT [116]. BRAFi-resistant melanoma cell lines displayed high active FAK levels [83], while a short-term (from 24 to 48 h) BRAFi treatment activated FAK in melanoma cells [39,47], which allowed for persistence under therapy through the acquisition of an NCSC phenotype [39]. Therefore, co-targeting FAK overcame BRAFi resistance [47,52]. Mechanistically, the FAK inhibitor reduced the proportion of NCSC persister cells, thus preventing the development of nongenetic (however, not genetic) resistance [39].

The membrane-tethered MT1-MMP was upregulated through TGF-β in BRAFi-resistant melanomas [117]. MT1-MMP1 remodelled the ECM and activated β1-integrin-FAK signalling to sustain survival under the MAPK blockade. Hence, direct MT1-MMP inhibition overcame resistance [117].

FAK signalling, along with matrix deposition and remodelling, can moreover be extrinsically triggered by BRAFi. Hirata et al. showed that BRAFi activated fibroblasts that, in turn, remodelled the ECM [52]. In agreement with these studies, the ECM-protective environment activated β1-integrin-FAK-SRC signalling in melanoma cells, which led to ERK reactivation and therapy resistance [52]. Furthermore, fibroblasts can be activated through the BRAFi-induced TGF-β secretion from melanoma cells. These activated fibroblasts increased the fibronectin deposition, HGF, and NRG, which promoted the survival of melanoma cells under therapy [118]. Therefore, the drug-resistant cells co-opted the stromal cells to generate a drug-protective environment [119].

## 4. Perspective and Unanswered Questions

In summary, the actomyosin cytoskeleton and many of its regulators are hyper-activated in MAPKi-resistant melanomas. This cytoskeletal remodelling occurs early on during therapy adaptation, rendering persister cells more capable of withstanding the MAPK blockade and later giving rise to completely therapy-resistant tumours. This cytoskeletal remodelling contributes to the ERK reactivation and/or provides parallel survival signalling. In addition, the cytoskeleton establishes a cross talk with the tumour’s microenvironment, which feeds back to tumour cells to sustain the tumour’s survival and, therefore, resistance (Figure 1).

However, this supposedly “enhanced” cytoskeleton [119] becomes an important vulnerability for therapy-resistant cells, which can be eradicated or at least can delay their emergence, by co-targeting certain cytoskeletal regulators along MAPKi. The cytoskeleton and its regulators are not commonly mutated in melanoma (except RAC1^P29S^) [12]. Therefore, targeting the cytoskeleton, which is essential for many physiological processes in non-tumour cells (cytokinesis, muscle contraction, and the migration of immune cells, among others) [79,81], could provoke undesirable toxicity. However, as with oncogenes (specifically, BRAF^V600E^), tumour cells (in particular, therapy-resistant cells) appear to be somehow “addicted” to the cytoskeletal’s remodelling signals, which would presumably lower the threshold for the effective targeting without affecting excessively normal cells.

However, there still remain some gaps in our knowledge. The precise mechanisms underlying the overactivation of cytoskeletal regulators in resistant cells are largely unknown. This is in part due to the fact that, except for in a few cases (YAP, MRTF, NMII, and EPHA2), the kinetics of cytoskeletal overactivation during therapy adaptation has not been assessed beyond the resistant vs. baseline state. In addition, it is unknown whether other cell cytoskeleton regulators (RHO GTPase GEFs or GAPs) or components from the other cell cytoskeletons, such as microtubules and other intermediate filaments, or septins, are, in addition, modulated in therapy-resistance.

Furthermore, apart from ROCK-NMII, do other cytoskeletal regulators involved in MAPKi resistance contribute to cross-resistance to ICB? In principle, some could, since they directly affect ECM remodelling and deposition. This at least serves as a physical barrier that could prevent the tumour’s infiltration by leukocytes (Figure 1). If immune function itself (i.e., immunosuppression) is controlled by other cytoskeletal regulators in the context of therapy has not been thoroughly investigated; however, there is evidence supporting this hypothesis. PD-L1 levels were higher in RAC1^P29S^ melanoma patients than those in wild-type RAC1 patients, and in fact, RAC1^P29S^ positively regulated PD-L1 protein levels in cultured cells [120] (Figure 2); however, the mechanism was not studied. In line with this, ROCK-mediated moesin phosphorylation regulated the PD-L1 stability by preventing its degradation in breast cancer cells [121]. In non-small cell lung cancer cells, TGF-β signaling upregulated the expression of MRTF (through a RhoA/ROCK non-canonical mechanism), which interacted with NF-κB/p65 to promote PD-L1 expression through transcription [122]. Whether these mechanisms occur in melanoma and their importance in the context of therapy resistance remain to be investigated.

Finally, it would be important to assess if cytoskeletal remodelling could be a vulnerability in other cutaneous melanoma subtypes (i.e., acral) or in non-cutaneous melanomas, such as uveal. There is evidence in favour of this hypothesis. Subsets of acral melanoma patients harbour amplifications of PAK1 (22% of tumours) and YAP1 (12% of tumours) loci [123], suggesting a dependency on these pathways for proliferation and/or invasion. Uveal melanomas driven by mutant GNAQ/GNA11 (Gαq proteins) activate FAK through TRIO-RHOA non-canonical Gαq; FAK, in turn, activates YAP, which promotes aberrant uveal melanoma growth [124]. Therefore, Gαq-FAK inhibition blocks xenograft growth [124]. Furthermore, co-targeting FAK and MEK impaired the growth of uveal melanoma xenografts and liver metastases [125].

## Figures and Tables

**Figure 1 cells-11-00518-f001:**
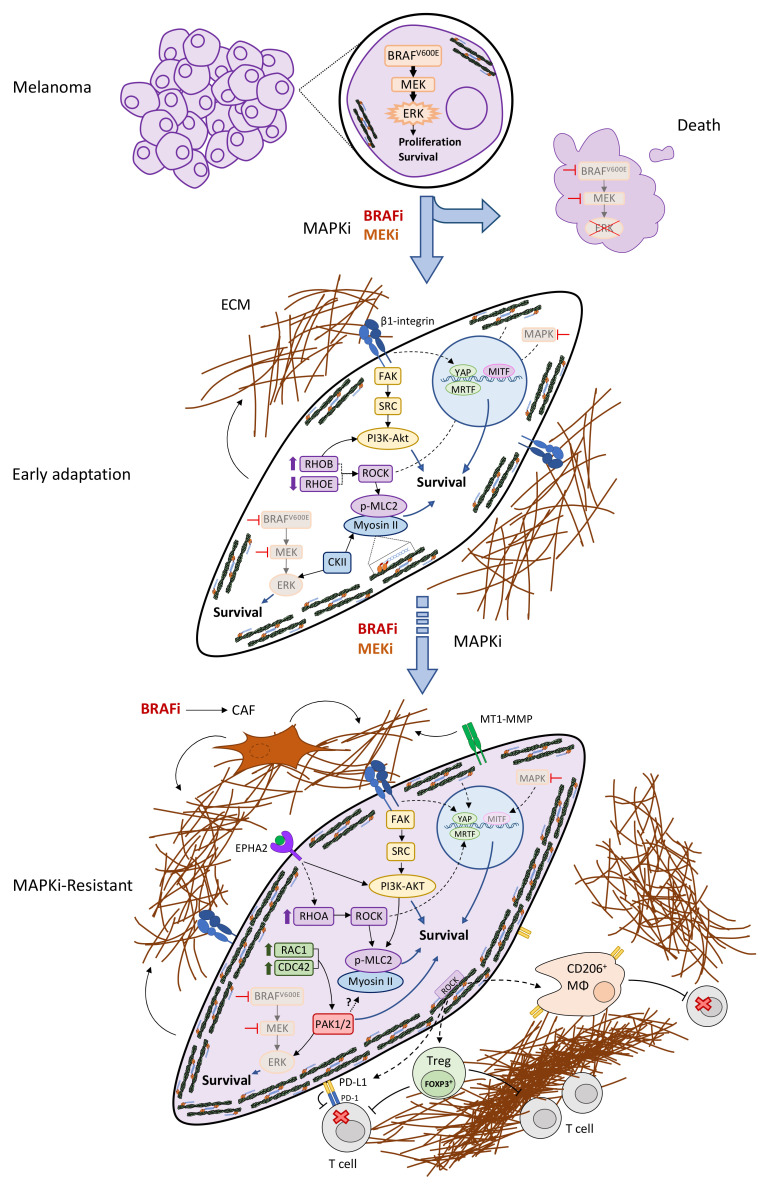
Cytoskeletal remodelling and hyperactivation during adaptation and development of resistance to therapy. BRAF^V600E^ increases ERK signalling to sustain aberrant growth and survival of melanoma cells. Upon MAPK-targeted therapy (MAPKi), most tumour cells rapidly die and tumours shrink, yet some are able to adapt and persist under drug pressure, displaying significant cytoskeletal remodelling. During early adaptation (24–72 h of treatment), enhanced signalling from RHO GTPases and effector proteins drives cytoskeletal remodelling through diverse, interconnected pathways. Most of these pathways play a crucial role in the development of therapy resistance and are later overactivated in MAPK-resistant melanoma cells (weeks or months of treatment). Signals from the ECM can also promote cytoskeletal reorganization; additionally, adapting and resistant cells can remodel the ECM and activate fibroblasts (CAFs (cancer-associated fibroblasts)), eventually creating a feed-forward mechanism and a drug-protective environment. BRAFi also promotes CAF generation, and hyperactive ROCK-NMII signalling generates an immunosuppressive tumour microenvironment (high PD-L1 on tumour cells, and high numbers of FOXP3^+^ Tregs and pro-tumourigenic CD206^+^ macrophages), which could mediate cross-resistance to immune checkpoint blockers in MAPKi-resistant tumours (MΦ, macrophage; Treg, regulatory T cell).

**Figure 2 cells-11-00518-f002:**
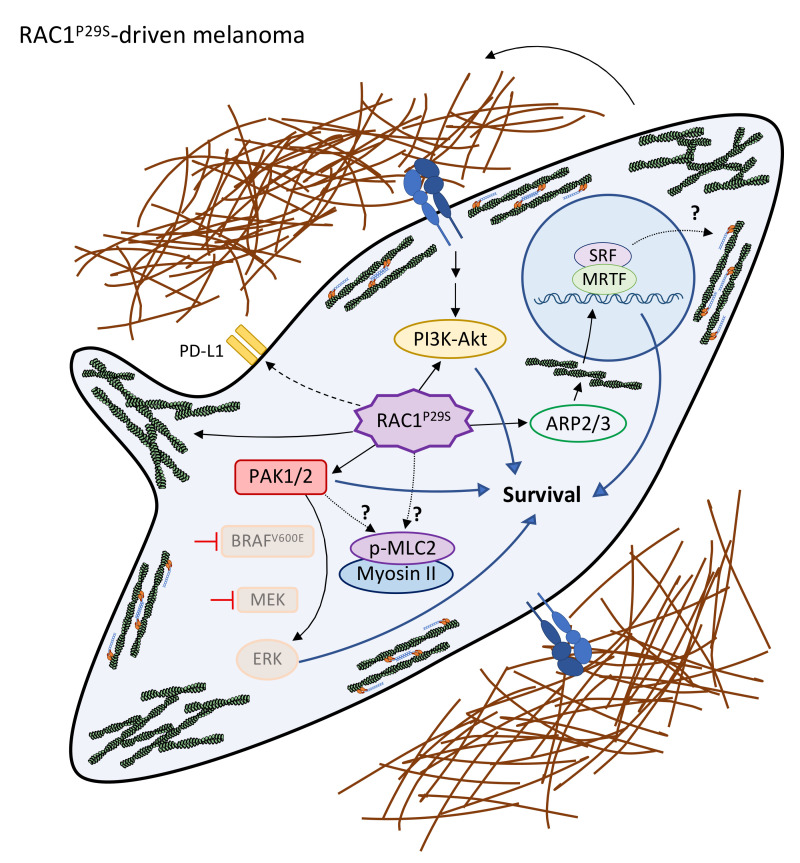
Cytoskeletal remodelling and hyperactivation in RAC1^P29S^-driven melanomas. RAC1^P29S^ and BRAF^V600E^ mutations frequently co-occur. Despite the efficacy of MAPK-targeted therapy, some cells can adapt and develop resistance, bypassing MAPK pathway blockade. Elevated RAC^P29S^ signalling plays a major role in this process, coordinating different downstream effectors. Some therapy resistance-promoting pathways are shared in BRAF^V600E^- and RAC1^P29S^-driven melanomas (integrin-PI3K-AKT), yet the roles of others, such as NMII activity, remain to be tested. RAC1^P29S^ also promotes the expression of PD-L1, suggesting a possible link to immune evasion and ICB resistance.

## Data Availability

Not applicable.

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
