# Peer review of "Cytoskeletal Remodelling as an Achilles’ Heel for Therapy Resistance in Melanoma"

_cells, 2022, doi:10.3390/cells11030518_

Round 1
Reviewer 1 Report
A narrative review on the cytoskeletal pathways involved in targeted therapy resistance and how these pathways could be applied to find new drugs to treat melanoma. I found some confusion in the introduction and in the future perspectives, but overall the article will be publishable after revisions:
The paper refers to melanoma in general and not only to cutaneous melanoma according to the title ; so paragraph 1 title and introduction should be changed accordingly; also, a sentence such as: "when interesting mucosal surfaces, melanoma tend to have even a worst prognosis" and a reference such as: doi: 10.3390/medicina57040359 should be added in the introduction.
In paragraph 4. Perspective and unanswered questions you speak about the involvement of cytoskeleton in acral melanoma, describing it as an entity different from cutaneous melanoma, which is not true, as acral melanoma is a subtype of cutaneous melanoma; in my opinion, you should say that the results reported in this paper, may not apply to all melanoma subtypes, such as...........
Saying that acral melanoma is not a cutaneous melanoma is wrong.
Thank You
.
Author Response
Review Report (Reviewer 1)
Comments and Suggestions for Authors.
We thank the reviewer for the constructive criticism that has helped improve our manuscript. Our responses are in blue.
A narrative review on the cytoskeletal pathways involved in targeted therapy resistance and how these pathways could be applied to find new drugs to treat melanoma. I found some confusion in the introduction and in the future perspectives, but overall the article will be publishable after revisions:
We have revised these sections and also added additional literature (cell of origin in melanoma and melanoma subtypes according to body location (lines 26-35), cross-resistance (lines 95-107)) that we hope has improved its readability.
The paper refers to melanoma in general and not only to cutaneous melanoma according to the title ; so paragraph 1 title and introduction should be changed accordingly; also, a sentence such as: "when interesting mucosal surfaces, melanoma tend to have even a worst prognosis" and a reference such as: doi: 10.3390/medicina57040359 should be added in the introduction.
We have changed the title of section 1 to “Melanoma and the problem of therapy resistance”, and have also clarified about the different locations where melanomas can arise. We have also added a more general reference on mucosal melanomas.
Lines 26-35: "Melanomas arise from transformation of melanocytes, which are neural-crest-derived cells specialized in the production of the pigment melanin that provides ultraviolet light protection [2]. Cutaneous melanoma originates from melanocytes in the basal layer of the epidermis, and it is the most common form of melanoma; therefore, the studies reviewed here mostly refer to cutaneous melanoma, unless otherwise stated. Other non-cutaneous melanocytes such as those in the choroidal layer of the eye, in gastrointestinal, respiratory and genitourinary mucosal surfaces, and in the meninges can undergo malignant transformation, albeit at much lower frequencies [2]. Some of these non-cutaneous melanomas such as mucosal melanomas have even a worst prognosis [3]. Melanocytes’ neural crest ancestry is thought to contribute to the high metastatic propensity of mela-nomas [4,5]."
In paragraph 4. Perspective and unanswered questions you speak about the involvement of cytoskeleton in acral melanoma, describing it as an entity different from cutaneous melanoma, which is not true, as acral melanoma is a subtype of cutaneous melanoma; in my opinion, you should say that the results reported in this paper, may not apply to all melanoma subtypes, such as........... Saying that acral melanoma is not a cutaneous melanoma is wrong.
We apologize for this oversight. We have corrected this mistake and added a reference to a study showing amplification of PAK1 and YAP1 in some acral melanomas (Yeh et al 2019 JNCI), suggesting potential dependencies on these cytoskeletal regulators.
Lines 460-469: “Finally, it would be important to assess if cytoskeletal remodelling could be a vulnerability in other cutaneous melanoma subtypes (i.e., acral), or in non-cutaneous melanomas, such as uveal. There is evidence in favour of this hypothesis. Subsets of acral melanoma patients harbour amplifications of PAK1 (22% tumours) and YAP1 (12% tumours) loci [123], suggesting a dependency on these pathways for proliferation and/or invasion. Uveal melanomas driven by mutant GNAQ/GNA11 (Gαq proteins) activate FAK through TRIO-RHOA non-canonical Gαq; FAK, in turn, activates YAP that promotes aberrant uveal melanoma growth [124]. Therefore, Gαq-FAK inhibition blocks xenograft growth [124]. Furthermore, co-targeting FAK and MEK impaired growth of uveal melanoma xenografts and liver metastases [125].”
Thank You
We thank again the reviewer for the constructive criticism that has helped improve our manuscript.
Reviewer 2 Report
This is an important comprehensive review regarding a potential role of the cytoskeleton in resistance to BRAF inhibitors. Figures are informative and of added value.
These are my comments for the review:
- The introduction and section 4 need thorough English editing. Some of the sentences are unclear.
- LINES 21-25 general introduction should include immunotherapy.
- LINES 26-31 in vast majority of melanomas- cell origin is melanocytic and cells are differentiated. The origin of melanocytes is neural crest. In a rare type – desmoplastic melanoma- cells acquire a different, spindle shape. Authors need to correct this paragraph.
- Lines 54-69 – There are inaccuracies in this paragraph. Specifically, most responders usually develop a durable response. Additionally, the “cross resistance” should be better discussed with additional literature. This is also referred to in several points along the manuscript, and should be clarified there as well.
- Section two – There are many reported mechanisms for resistance to BRAFi. This paragraph is not well organized and additional data is required.
- The role of cytoskeleton in response to immunotherapy should be more detailed in the last paragraph of open questions, and should include works already published in that area. For example – RAC1 mutation was shown to correlate with high PDL1 . this should be also added to figure 2
( Vu HL, Rosenbaum S, Purwin TJ, Davies MA, Aplin AE. RAC1 P29S regulates PD-L1 expression in melanoma. Pigment Cell Melanoma Res 2015; 28:590-8; PMID:26176707; http://dx.doi.org/10.1111/pcmr.12392 [PMC free article] [PubMed] [CrossRef] [Google Scholar])
Minor comments
- The authors use the words “kill the cells” along the manuscript. This should be replaced with cell death/ tumor shrinkage
- Line 23-24 – should be “is curative”
Author Response
Review Report (Reviewer 2)
Comments and Suggestions for Authors
This is an important comprehensive review regarding a potential role of the cytoskeleton in resistance to BRAF inhibitors. Figures are informative and of added value.
We thank the reviewer for the kind comments on our manuscript and for the constructive criticism, which has helped improve our manuscript. Our responses are in blue.
These are my comments for the review:
- The introduction and section 4 need thorough English editing. Some of the sentences are unclear.
We have revised these sections and also added additional literature (cell of origin in melanoma and melanoma subtypes according to body location (lines 26-35), cross-resistance (lines 95-107)) that we hope has improved its readability.
- LINES 21-25 general introduction should include immunotherapy.
We have changed this paragraph including traditional chemotherapies (dacarbazine) and immunotherapies (IL2) as follows (lines 21-25)
“Melanoma is a highly aggressive cancer with a high propensity to metastasize. Early diagnosis and surgical removal of local, non-invasive tumours is curative in most cases [1]. However, late diagnosis of metastatic non-resectable melanomas carries a poor prognosis, since traditional chemotherapies (dacarbazine) or immunotherapies (high dose interleukin 2) are not effective in most cases [1].”
- LINES 26-31 in vast majority of melanomas- cell origin is melanocytic and cells are differentiated. The origin of melanocytes is neural crest. In a rare type – desmoplastic melanoma- cells acquire a different, spindle shape. Authors need to correct this paragraph.
We have added a new paragraph describing the origin of melanomas and melanoma subtypes according to location, lines 26-35:
“Melanomas arise from transformation of melanocytes, which are neural-crest-derived cells specialized in the production of the pigment melanin that provides ultraviolet light protection [2]. Cutaneous melanoma originates from melanocytes in the basal layer of the epidermis, and it is the most common form of melanoma; therefore, the studies reviewed here mostly refer to cutaneous melanoma, unless otherwise stated. Other non-cutaneous melanocytes such as those in the choroidal layer of the eye, in gastrointestinal, respiratory and genitourinary mucosal surfaces, and in the meninges can undergo malignant transformation, albeit at much lower frequencies [2]. Some of these non-cutaneous melanomas such as mucosal melanomas have even a worst prognosis [3]. Melanocytes’ neural crest ancestry is thought to contribute to the high metastatic propensity of melanomas [4,5].”
We have also revised the paragraph about phenotypes, lines 36-64
“Melanoma cells display enormous plasticity at epigenetic, transcriptional, translational and post-translational levels [6]. Cutaneous melanoma cell lines and human tumours can be classified, according to their gene expression profile, into 4 phenotypes characterized by different degrees of differentiation and invasive capabilities: undifferentiated, neural crest-like, transitory, melanocytic [7–9]. The majority of human melanomas from The Cancer Genome Atlas (TCGA) database can be classified as differentiated (melanocytic and transitory phenotypes) [8]. However, single-cell analysis shows that tumours can be composed of malignant cells of different phenotypes [9], strongly supporting the occurrence of phenotype interconversion [7–10]. This phenotypic plasticity, or phenotype switching, is thought to greatly contribute not only to tumour progression, dissemination and metastasis, but also, as discussed below, to therapy failure [10].”
- Lines 54-69 – There are inaccuracies in this paragraph. Specifically, most responders usually develop a durable response.
We have corrected this text in current lines 91-92 to “However, the majority of eligible melanoma patients do not respond to these therapies due to primary resistance [29].”
Additionally, the “cross resistance” should be better discussed with additional literature. This is also referred to in several points along the manuscript, and should be clarified there as well.
We have revised this part adding more supporting literature as follows:
Lines 95-107: “MAPK inhibitor (MAPKi)-resistant tumours harbour transcriptional alterations in signalling pathways related to invasion and metastasis (epithelial-to-mesenchymal transition, TGF-beta, ECM remodelling) and also immunosuppression (T cell deficiency/ exhaustion and loss of antigen presentation), among other alterations [30], suggesting cross-resistance to salvage immune checkpoint blockers (ICB). This is supported by transcriptional analyses of tumours from ICB-non-responders, which showed alterations in similar invasion/metastasis pathways [31]. In fact, some retrospective studies suggest that progression to MAPKi is associated with inferior responses to subsequent ICB treatment [32–34]. Mechanistically, BRAFi-resistant tumours are in fact cross-resistant to ICB due to ERK reactivation and subsequent enhanced MAPK pathway transcriptional output, which establishes an immunosuppressive environment with dysfunctional dendritic cells [35]. Therefore, preclinical studies have shown that lead-in treatment with ICB before MAPKi maximizes antitumour immunity and efficacy [36,37].”
Lines 290-299: “Importantly, ROCK-NMII also altered the tumour microenvironment by establishing immunosuppression (increased pro-tumorigenic CD206+ macrophages and FOXP3+ Tregs), suggesting a contribution to cross-resistance to ICB (anti-PD-1) [68]. In fact, ROCKi combined with anti-PD-1 decreased growth of anti-PD-1 resistant melanomas via both intrinsic tumour cell death and relief of immunosuppression, without affecting tumour infiltration by CD4+ or CD8+ T cells [68]. ROCKi also reduced PD-L1 (PD-1 ligand) levels in both tumour cells and CD206+ macrophages [68], which could contribute to reduce immunosuppression and overcoming ICB resistance. STAT3 promotes PD-L1 expression [102,103], so ROCKi could be reducing PD-L1 levels via blockade of the ROCK-STAT3 cross-talk [89].”
Lines 440-441: “Furthermore, apart from ROCK-NMII, do other cytoskeletal regulators involved in MAPKi resistance also contribute to cross-resistance to ICB?”
- Section two – There are many reported mechanisms for resistance to BRAFi. This paragraph is not well organized and additional data is required.
We have added an additional paragraphon resistance mechanisms, along with a list of more comprehensive reviews for interested readers:
Lines 153-166: “Regarding the resistant state, most resistant tumours (70%) display MAPK pathway reactivation through a variety of mechanisms [16]: 1) drug target alterations through genetic mechanisms (BRAF amplification [49,50] and alternative splicing [51]; 2) hyperactivation of compensatory signalling pathways (FAK [52], SRC [53], PI3K-AKT [54] and STAT3 [55]) via upregulation of receptor tyrosine kinases (AXL [56], IGFR1 [18], MET [57], EGFR [53,58], PDGFRB [59]), along with loss of tumour suppressor PTEN [60], and activating RAS mutations [30]; 3) activation of parallel effector COT/MAP3K8 [61]; 4) activation of downstream effectors (activating MEK mutations [49,62]). In addition, a few resistant tumours do not restore ERK phosphorylation [49,54,62,63], but seem to bypass the MAPK pathway through activation of more downstream oncogenic output via GPCR-cAMP-CREB signalling and amplification of MITF [64]. In addition, intrinsically resistant tumours seem to be in an alternative transcriptional state/phenotype characterized by activation of NF-kB and AXL signalling and low MITF activity) [65,66]. More comprehensive reviews on resistance to MAPKi in melanoma can be found elsewhere [6,16,38,67].”
- The role of cytoskeleton in response to immunotherapy should be more detailed in the last paragraph of open questions, and should include works already published in that area. For example – RAC1 mutation was shown to correlate with high PDL1 . this should be also added to figure 2 ( Vu HL, Rosenbaum S, Purwin TJ, Davies MA, Aplin AE. RAC1 P29S regulates PD-L1 expression in melanoma. Pigment Cell Melanoma Res2015; 28:590-8; PMID:26176707; http://dx.doi.org/10.1111/pcmr.12392 [PMC free article] [PubMed] [CrossRef] [Google Scholar])
We thank the reviewer for the suggestion. We have added the reference on RAC1P29S and PD-L1, and other papers showing similar links in other types of cancer:
Lines 440-454: "Furthermore, apart from ROCK-NMII, do other cytoskeletal regulators involved in MAPKi resistance also contribute to cross-resistance to ICB? In principle, some of them could, since they directly affect ECM remodelling and deposition, which at least serves as a physical barrier that could prevent tumour infiltration by leukocytes (Figure 1). If immune function itself (i.e., immunosuppression) is controlled by other cytoskeletal regulators in the context of therapy has not been thoroughly investigated, but there is evidence supporting this hypothesis. PD-L1 levels were higher in RAC1P29S melanoma patients vs wild-type RAC1 patients, and in fact RAC1P29S positively regulated PD-L1 protein levels in cultured cells [120], although the mechanism was not studied. In line with this, ROCK-mediated moesin phosphorylation regulated PD-L1 stability by preventing its degradation in breast cancer cells [121]. In non-small cell lung cancer cells TGF-β signaling upregulated the expression of MRTF (through a RhoA/ROCK non-canonical mechanism), which interacted with NF-κB/p65 to promote PD-L1 expression via transcription [122]. Whether these mechanisms occur in melanoma and their importance in the context of therapy resistance remains to be investigated.”
We have also added a link between RAC1P29S and PD-L1 in Figure 2.
Lines 262-264: “RAC1P29S also promotes expression of PD-L1, suggesting a possible link to immune evasion and ICB resistance.”
We have also revised Figure 1 legend to include involvement of ROCK-NMII in sustaining PD-L1 levels and immunosuppressive populations, since although this was already in Figure 1 in the original submission, it was not mentioned in the legend.
Lines 234-237: “In addition, hyperactive ROCK-NMII signalling generates an immunosuppressive tumour microenvironment (high PD-L1 on tumour cells, high numbers of FOXP3+ Tregs and pro-tumourigenic CD206+ macrophages) that could mediate cross-resistance to immune checkpoint blockers in MAPKi-resistant tumours. MΦ, macrophage. Treg, regulatory T cell.”
Minor comments
- The authors use the words “kill the cells” along the manuscript. This should be replaced with cell death/ tumor shrinkage
We have replaced it to cell death/shrinkage/survive/eradicate as per reviewer’s suggestion (lines 70, 294, 416).
- Line 23-24 – should be “is curative”
We have changed this sentence (lines 21-22) to “Early diagnosis and surgical removal of local, non-invasive tumours is curative in most cases [1].”